

# Tracking 'transitional' diadectomorphs in the earliest Permian of equatorial Pangea

Gabriela Calábková[1,2], Daniel Madzia[3], Vojtěch Nosek[4] and Martin Ivanov[2]

[1] Department of Geology and Paleontology, Moravian Museum, Brno, Czech Republic
[2] Department of Geological Sciences, Faculty of Science, Masaryk University, Brno, Czech Republic
[3] Department of Evolutionary Paleobiology, Institute of Paleobiology, Polish Academy of Sciences, Warsaw, Poland
[4] Department of Archaeology and Museology, Faculty of Arts, Masaryk University, Brno, Czech Republic

## ABSTRACT

Diadectomorpha was a clade of large-bodied stem-amniotes or possibly early-diverging synapsids that established a successful dynasty of late Carboniferous to late Permian high-fiber herbivores. Aside from their fairly rich record of body fossils, diadectomorphs are also well-known from widely distributed tracks and trackways referred to as *Ichniotherium*. Here, we provide detailed description of a diadectomorph trackway and a manus-pes couple originating from two different horizons in the Asselian (lowermost Permian) of the Boskovice Basin in the Czech Republic. The specimens represent two distinct ichnotaxa of *Ichniotherium*, *I. cottae* and *I. sphaerodactylum*. Intriguingly, the *I. cottae* trackway described herein illustrates a 'transitional' stage in the posture evolution of diadectomorphs, showing track morphologies possibly attributable to a *Diadectes*-like taxon combined with distances between the successive manus and pes imprints similar to those observable in earlier-diverging diadectomorphs, such as *Orobates*. In addition, this trackway is composed of 14 tracks, including six well-preserved manus-pes couples, and thus represents the most complete record of *Ichniotherium cottae* described to date from the Asselian strata. In turn, the manus-pes couple, attributed here to *I. sphaerodactylum*, represents only the second record of this ichnotaxon from the European part of Pangea. Our study adds to the diversity of the ichnological record of Permian tetrapods in the Boskovice Basin which had been essentially unexplored until very recently.

## INTRODUCTION

Diadectomorpha was a widely distributed clade of large-bodied stem-amniotes (*e.g.*, *Panchen & Smithson, 1988*; *Laurin & Reisz, 1997*, *1999*; *Pardo et al., 2017*; *Ford & Benson, 2020*; *Brocklehurst, Ford & Benson, 2022*) or perhaps early-diverging synapsids (*Berman, 2000*, *2013*; *Klembara et al., 2019*; *Klembara et al., 2021*; *Clack, Smithson & Ruta, 2022*; see also the phylogenetic assessment of *Marjanović & Laurin (2019)*) that originated in the Carboniferous (see, *e.g.*, *Voigt & Ganzelewski, 2010*), flourished in late Pennsylvanian and

Corresponding author
Gabriela Calábková,
g.calabkova@gmail.com

Cisuralian (late Carboniferous and early Permian; *e.g.*, *Berman & Sumida, 1990*; *Berman, Sumida & Lombard, 1992*; *Berman, Sumida & Martens, 1998*; *Berman et al., 2004*), and died out in or shortly after the Wuchiapingian (late Permian; *Liu & Bever, 2015*). Owing to their phylogenetic placement and recognition as one of the earliest tetrapod lineages to evolve high-fiber herbivory (*e.g.*, *Beerbower, Olson & Hotton, 1992*; *Hotton, Olson & Beerbower, 1997*; *Sues, 2000*), diadectomorphs are significant contributors to our understanding of the amniote origins and the structure of land ecosystems in the late Paleozoic.

Asides from their fairly abundant body-fossil record, diadectomorphs are also well-known from their fossil tracks and trackways. Three distinct ichnotaxa associated with diadectomorph trackmakers are currently distinguished: *Ichniotherium praesidentis*, *Ichniotherium sphaerodactylum*, and *Ichniotherium cottae*. Among these, *I. praesidentis* is the oldest and rarest morphotype, being only known from the Westphalian A (Moscovian, mid-Pennsylvanian) of the Bochum Formation, Germany (*Voigt & Ganzelewski, 2010*); *I. sphaerodactylum* has been reported from the Gzhelian (uppermost Carboniferous) to Artinskian (lower Permian) of Arizona (*Francischini et al., 2019*), Canada (*Brink, Hawthorn & Evans, 2012*), Germany (*Voigt, Berman & Henrici, 2007*; *Voigt & Haubold, 2000*; *Marchetti, Voigt & Santi, 2018*; *Calábková, Ekrt & Voigt, 2023*), and Morocco (*Voigt et al., 2011*); and *I. cottae*, the most common and abundant of the ichnotaxa, is known from the Moscovian (upper Carboniferous) to Artinskian of the Czech Republic (*von Hochstetter, 1868*; *Frič, 1887*; *Calábková & Nosek, 2022*), France (*Mujal & Marchetti, 2020*), Germany (*Voigt & Haubold, 2000*; *Voigt, 2005*), Great Britain (*Haubold & Sarjeant, 1974*), Morocco (*Lagnaoui et al., 2018*), Poland (*Voigt et al., 2012*), and Colorado (*Voigt, Small & Sanders, 2005*), New Mexico (*Voigt & Lucas, 2015*), and Ohio (*Baird, 1952*) in the United States.

Here, we describe a trackway and a manus-pes couple representing two diadectomorph ichnotaxa, *I. cottae* and *I. sphaerodactylum*, respectively. No body-fossil remains of diadectomorphs have been discovered in the Czech Republic so far. However, the presence of tracks ascribed to *I. cottae* has previously been mentioned to derive from two units in the Czech Republic, including the Boskovice Basin (*von Hochstetter, 1868*; *Calábková & Nosek, 2022*) and the Krkonoše Piedmont Basin (*Frič, 1887*). Nevertheless, these reports were brief and did not assess the material in detail. In turn, *I. sphaerodactylum* has not been described from the Czech Republic before and, in fact, represents only the second record of this ichnotaxon from the European part of Pangea, the first being from Germany (*Voigt, Berman & Henrici, 2007*).

The material described herein originates from two localities, Čebín and Zbýšov, situated at different horizons in the Asselian (lowermost Permian) of the Boskovice Basin. From the viewpoint of tetrapod fossil record, the lowermost Permian strata of the Boskovice Basin have long been renowned for materials of taxa inhabiting aquatic environment, including extraordinarily abundant specimens of discosauriscid seymouriamorphs (*e.g.*, *Špinar, 1952*; *Klembara, 1995*, *2005*, *2016*) and rare temnospondyls (*Augusta, 1947*; *Milner, Klembara & Dostál, 2007*; *Klembara & Steyer, 2012*; *Werneburg et al., 2023*). However, recent fieldwork conducted in the basinal strata at several localities provided diverse

assemblages of tetrapod footprints which fundamentally enriches our knowledge of the tetrapod biodiversity in the Permian terrestrial settings in this area, revealing the presence of large-bodied seymouriamorphs (*Calábková, Březina & Madzia, 2022*), early-diverging synapsids (*Calábková et al., 2023*), and now two distinct diadectomorphs.

We provide detailed description of the *Ichniotherium* material from the lowermost Permian of the Boskovice Basin, illustrate it through image-based modeling, and assess its potential trackmakers' affinities using multivariate analyses.

## Geological setting

The Boskovice Basin represents a NNE–SSW-oriented half-graben situated in the eastern margin of the Bohemian Massif that is about 100 km long and 3–10 km wide. The basin was part of the Variscan orogenic belt in the equatorial Pangea. Sedimentation started in the southern part of the basin during the Gzhelian (latest Carboniferous) and continued uninterrupted towards to north through the early Permian (*Jaroš, 1963*; *Jaroš & Malý, 2001*; *Pešek, 2004*). The marginal facies are composed of the Balinka conglomerates in the west and the Rokytná conglomerates in the east that are interpreted as residues of an alluvial fan system that prograded towards the basin, diachronously with the sedimentation of all formations (*e.g.*, *Jaroš, 1962*; *Houzar et al., 2017*). The intrabasinal sedimentary complex is composed of cyclically arranged fluvial to fluvio-lacustrine clastic deposits, mostly red-colored, with the co-occurrence of grey-colored units indicating short-term semi-humid oscillations. The grey clastics are mostly represented by lacustrine horizons bearing a rich fossil record (*Jaroš, 1963*; *Pešek, 2004*; *Opluštil et al., 2017*). The cumulative thickness of the deposits is estimated to have been up to 5–6 km thick (*Jaroš, 1963*; *Jaroš & Malý, 2001*; *Pešek, 2004*), whereas seismic data indicate around 3 km (*Dopita, Havlena & Pešek, 1985*).

The depocenter of the basin is divided into southern and northern sub-basins, separated by the Tišnov-Kuřim Elevation; Rosice Oslavany Sub-basin on the southern part and Letovice Sub-basin on the northern part (*Havlena, 1964*; *Jaroš & Malý, 2001*; *Pešek, 2004*). The older Rosice-Oslavany sub-basin is divided into the Rosice-Oslavany and Padochov formations and the younger Letovice sub-basin comprises the Veverská Bítýška and Letovice formations (*Jaroš & Malý, 2001*; *Pešek, 2004*).

The specimen MZM Ge33302 is preserved in fine-grained sandy floodplain deposits discovered at Zbýšov, which is situated within sandstone beds between the Zbýšov and Říčany horizons in the middle part of the upper section of the Padochov Formation (Asselian) (*Jaroš & Malý, 2001*). The specimen PM PAL113, preserved in fine-grained sandy floodplain deposits, originates from Čebín, a locality that is situated within red clastic sediments deposited approximately at the level of the Chudčice Horizon in the uppermost part of the Veverská Bítýška Formation (Asselian) (*Jaroš & Malý, 2001*) on the very border with the Letovice Formation (Fig. 1).

The faunal components of the Padochov Formation consist of bivalves, clam shrimps ('conchostracans'), insects, acanthodians, xenacanthids, branchiosaurids, indeterminate, very rare finds of discosauriscids, and tetrapod fossil tracks (*e.g.*, *Kukalová, 1959*, *1965*; *Schneider, 1984*; *Ivanov, 2003*; *Milner, Klembara & Dostál, 2007*; *Štamberg & Zajíc, 2008*;

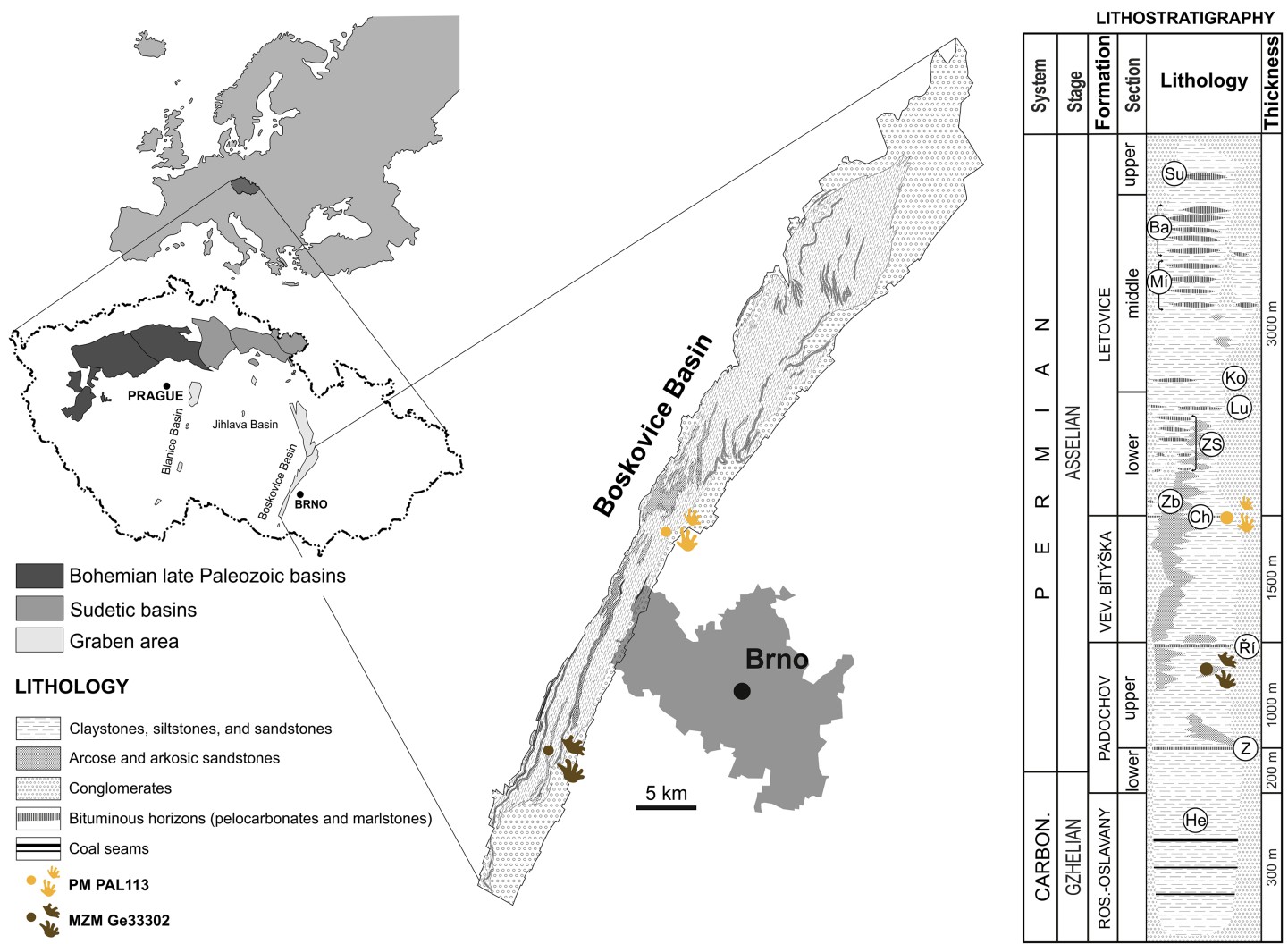

**Figure 1 Location of the studied area with a lithostratigraphic scheme.** Horizons: He, Helmhacker; Z, Zbýšov; Ř, Říčany; Ch, Chudčice; Zb, Zbraslavec; ZS, Zboněk-Svitávka; Lu, Luběm; Ko, Kochov; Mí, Míchov; Ba, Bačov; Su, Sudice. Lithostratigraphy follows *Jaroš & Malý (2001)*, *Pešek (2004)*, and *Zajíc & Štamberg (2004)*. Chronostratigraphy follows *Štamberg (2014)* and J. Jirásek, 2021, personal communication. Figure is modified from *Calábková et al. (2023)*.

*Schneider & Werneburg, 2012*; *Štamberg, 2014*; *Calábková, Březina & Madzia, 2022*; *Calábková et al., 2023*). No fossil fauna has been recorded from the Veverská Bítýška Formation. The diadectomorph footprints from Čebín, therefore, provide the first evidence of the presence of faunal components in this formation. The fauna of the Letovice Formation contains bivalves, clam shrimps ('conchostracans'), insects, acanthodians, actinopterygians, temnospondyls, discosauriscids and tetrapod fossil tracks (*e.g.*, *Kukalová, 1963*, *1964*; *Schneider, 1980*, *1984*; *Klembara, 1997*, *2005*, *2009*; *Schneider & Werneburg, 2006*, *2012*; *Štamberg, 2007*, *2014*; *Klembara & Steyer, 2012*; *Klembara & Mikudíková, 2018*; *Werneburg et al., 2023*; *Calábková, Březina & Madzia, 2022*; *Calábková et al., 2023*).

## MATERIALS AND METHODS

### Material

This study is based on two specimens: PM PAL113 (Figs. 2A–2E), housed at the collections of the Podhorácké Museum in Předklášteří, Czech Republic, and MZM Ge33302 (Figs. 3A–3C), deposited in the Moravian Museum in Brno, Czech Republic. The specimen PM PAL113 comprises a trackway of *Ichniotherium cottae* bearing 14 tracks crossed by desiccation cracks. In turn, the specimen MZM Ge33302 comprises a manus-pes couple attributable to *Ichniotherium sphaerodactylum*.

PM PAL113 has been part of the paleontological collections of the Podhorácké Museum in Předklášteří since the first half of the 20th century (*Šmarda, 1931*); MZM Ge33302 was found by Tomáš Viktorýn during fieldwork conducted in 2022. Both samples are preserved as convex hyporelief.

### Anatomical terminology and measurements

The anatomical terminology and protocol for obtaining measurements follow those of *Leonardi (1987)* and *Buchwitz & Voigt (2018)*. Measurements were obtained using a digital caliper and ImageJ. The track and trackway measurements of PM PAL113 were calculated omitting the desiccation cracks to avoid distortion of the track and trackway parameters.

### Multivariate analyses

In order to reconstruct the morphospace occupation of PM PAL113 among diadectomorph trackways we utilized the parameters published by *Buchwitz et al. (2021*; Supplementary Table 2), *Buchwitz & Voigt (2018*; average values from measured step cycles of specimen Kletno No.1 and Marieta_NA; supplementary 3 and 4), and *Mujal & Marchetti (2020*; average values of trackway 1 of MNHN-LOD 83; table 1 and 2), added data obtained from PM PAL113, and performed a principal component analysis (PCA) using PAST 4.12b (*Hammer, Harper & Ryan, 2001*). Prior to the analysis, all raw continuous variables were z-transformed. The original values and z-scores, and the extended results of the PCA, are provided in Material SI. A .dat file, executable in PAST, is provided in Material SII. The same .dat file was used to construct box plots with selected values.

### Image-based modeling

Our protocol for three-dimensional (3D) modeling follows *Porter, Roussel & Soressi (2016)*. In order to fully cover the surface of the samples, we obtained 100 images of each specimen, PM PAL113 and MZM Ge33302, in two elevation positions. The photos were taken using a full frame camera Nikon D750 (lens Tamron 24–75 mm, F2.8). The images were subsequently processed to reconstruct 3D photogrammetric models using the software Agisoft Metashape PRO 1.8. The procedure was complemented with scans using the geo-referenced marker grid matrix. This approach results in a greater precision than geo-referencing of models through one or more scale bars. The models have been reconstructed in the highest possible quality (3.5 million polygons), and visualized and interpreted through CloudCompare 2.10 and Blender 3.0.

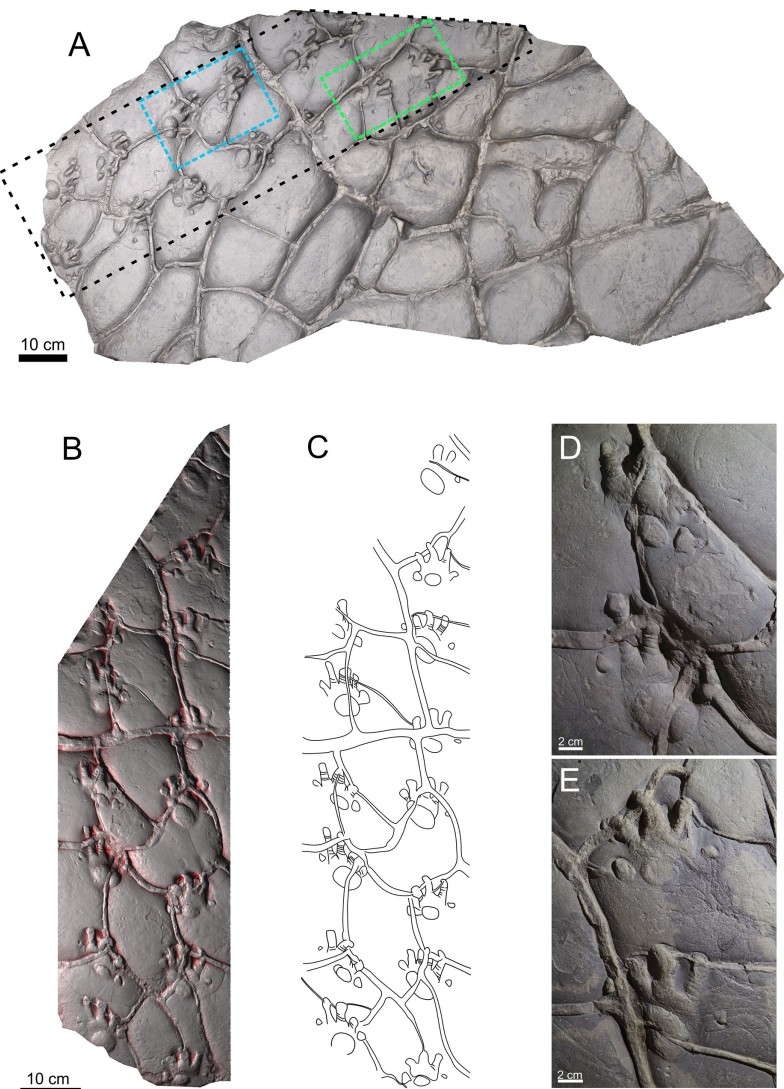

**Figure 2 Trackway of *Ichniotherium cottae*, PM PAL113, convex hyporelief.** Image-based modeling of whole slab (A). The black dashed line depicted the trackway (B) and outline drawing (C). The blue and green dashed lines show the manus-pes couples (D, E).

The meshes obtained from physical samples using the Structure from Motion method are available through the MorphoSource data archive: https://www.morphosource.org/projects/000546695.

## RESULTS

### Systematic paleoichnology

*Ichniotherium* Pohlig, 1892
*Ichniotherium cottae* (Pohlig, 1885)

**Material.** PM PAL113, a trackway composed of 14 tracks and including six manus-pes couples (Figs. 2A–2E).

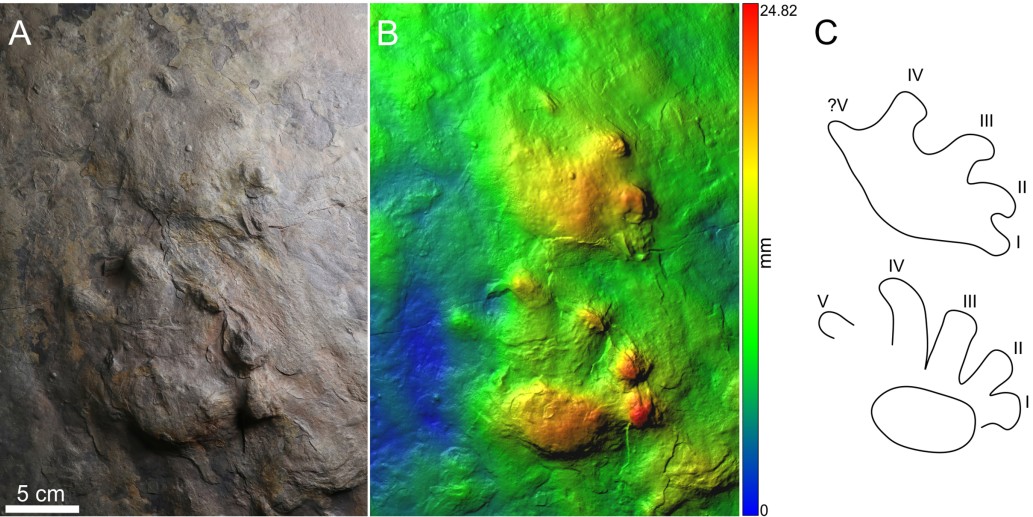

**Figure 3 Manus-pes couple of *Ichniotherium sphaerodactylum*, MZM Ge33302, convex hyporelief.** Photo (A), image-based modeling (B), and outline drawing (C).

**Locality and horizon.** Čebín, South Moravian Region, Czech Republic; most likely uppermost part of the Veverská Bítýška Formation (Fig. 1), Asselian, lowermost Permian, Boskovice Basin.

## Description and comparisons

The manus and pes imprints are plantigrade and pentadactyl. The pes imprints (104.6 mm long) are larger than the manus imprints (76.08 mm long; Material SI). The pes imprints are as wide as long, whereas the manus imprint is slightly wider than long. The pedal digit imprints are rather straight, and the manual digits II–IV are often slightly bent inwardly. The digit imprints show typical rounded "drumstick-like" terminations. Flexion creases are often visible on the impressions of the digits. In both, the manus and pes imprints, the digit lengths increase from digit I to IV, and the digits V are slightly shorter or the same sized as digits II. The pV/pIV ratio is 0.60 in average. The palm and sole impressions are wider than long and form elliptical to subcircular shapes. The palm impression lies mostly opposite to digits II–III in the manus, whereas the sole impression usually lies opposite to digits II–IV. The tracks show the medial-median functional prevalence. The trackway shows an alternating arrangement of successive manus and pes imprints. The overstepping does not occur. The particular trackway measurements (in average values) include: parallel to slightly outward rotation of the pedal imprints (−2.1°) and parallel or slightly inward rotation of the manual imprints (5°), manual pace angulations is 83.5°, pedal pace angulation is 87.4°, pedal stride length/pes length is 3.06, pedal gauge width/pes length is 1.49, pedal pace length/pes length is 2.15, manus-pes distance/pes length ratio is 1.27 (Material SI). All footprints are crossed by desiccation cracks formed after tracks registration.

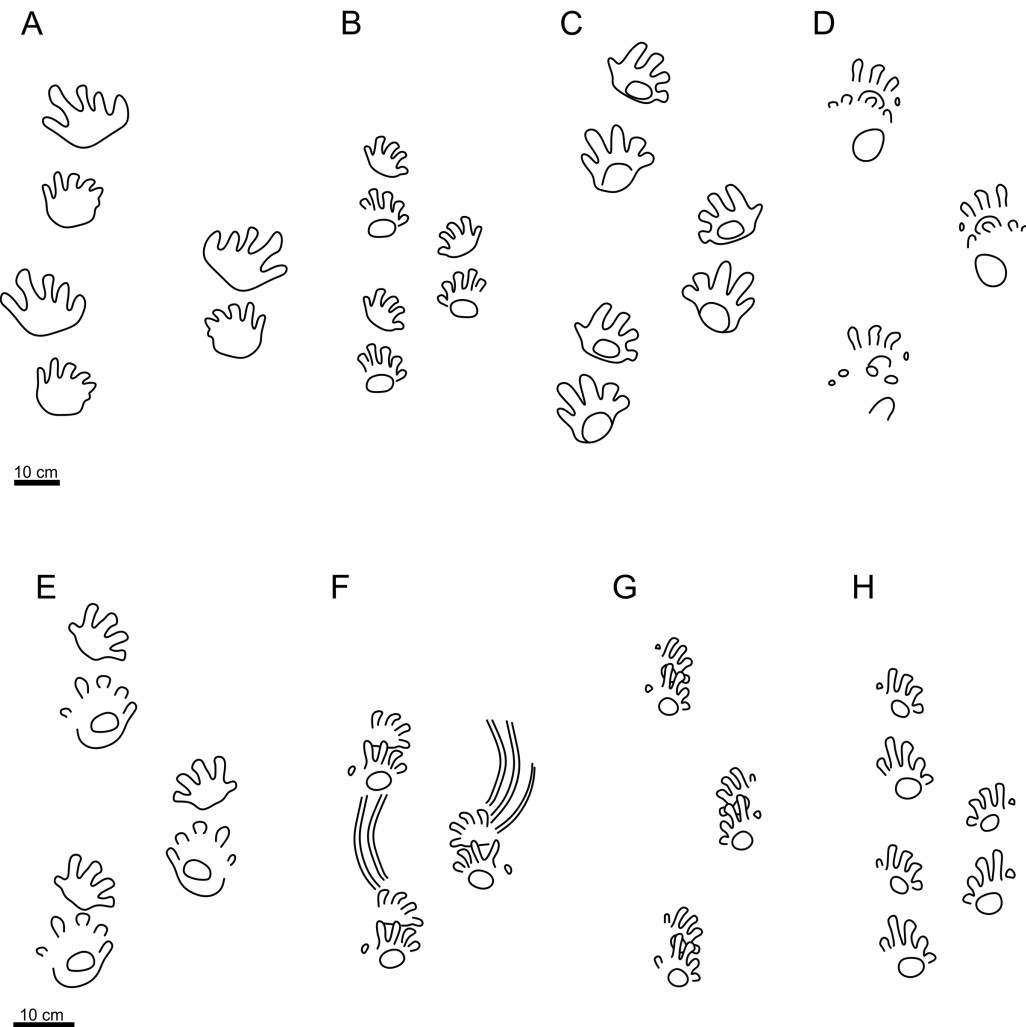

**Figure 4** Schematic trackway pattern of *Ichniotherium praesidentis, Ichniotherium sphaerodactylum* and *Ichniotherium cottae*. Schematic trackway pattern of *Ichniotherium praesidentis* (A), *Ichniotherium sphaerodactylum* (B) and various morphotypes of *Ichniotherium cottae*; "Hainesi-Willsi type" (C, D), "Gottlob-Birkheide type" (E, F), "Bromacker type" (G), and PM PAL113 (H). Illustrations of individual patterns follow *Voigt & Ganzelewski (2010)* and *Buchwitz & Voigt (2018)*.

## Remarks

The specimen PM PAL113 shows all diagnostic features of *I. cottae*, such as relatively short pedal digit V with pV/pIV ratio = 0.60 (in average value), and the palm and the sole impressions forming elliptical to subcircular shapes. The ichnospecies *Ichniotherium praesidentis* differs from PM PAL113 because of a longer pedal digit V impression corresponding to the length of the pedal digit III impression, a prominent manual basal pad I impression, an inversed alternation of the pattern of manus-pes couples, a more acute pace angulation, a lower stride length/pes length ratio, and a strong outward rotation of the pes imprints (Fig. 4A; Material SI). The ichnospecies *Ichniotherium sphaerodactylum* differs from PM PAL113 in a distinctly longer pedal digit V impressions which are subequal to or even longer than the length of pedal digit III impression (pV/pIV

> 0.60), the palm impressions are usually not clearly delimited, the sole impressions are often wider and lie opposite to digits II–V. Furthermore, the trackway pattern of *I. sphaerodactylum* usually shows a lower stride length/pes length ratio and a more acute pace angulation (Fig. 4B; Material SI). Other early Permian ichnotaxa reaching size similar to that of *Ichniotherium* include *Dimetropus* and *Limnopus*. *Dimetropus* can be clearly distinguished from PM PAL113 based on its typical proximodistally extended palm/sole impressions, deeply impressed metapodial-phalangeal pads, and relatively short and straight digit imprints with deeply impressed clawed terminations. In turn, *Limnopus* differs from PM PAL113 in having a tetradactyl manus imprint with short, deeply impressed digits which are often not separated from the palm impression (see, *e.g.*, *Voigt, 2005*).

    *Ichniotherium sphaerodactylum* (Pabst, 1895)

**Material.** MZM Ge33302, a manus-pes couple (Figs. 3A–3C).

**Locality and horizon.** Zbýšov, South Moravian Region, Czech Republic; upper section of the Padochov Formation (Fig. 1), Asselian, lowermost Permian, Boskovice Basin.

## Description and comparisons

The manus and pes imprints are plantigrade and pentadactyl. The pes imprints are larger (121.5 mm long; Material SIII) than the manus imprints (93.5 mm long). The manus and the pes imprints are wider than long, while the pes imprints are only slightly wider. The pedal digits I–III are straight, whereas distal portions of the pedal digits IV–V are bent outwardly. The manual digits II–III are bent inwardly. The digits show typical rounded "drumstick-like" terminations. The digit length increases from digit I to IV, the pedal digit V is approximately as long as the pedal digit III. The manual digit V is poorly preserved or not preserved at all. The pV/pIV ratio is 0.88. The palm and the sole impressions are broad and elliptical in shape. The tracks show the medial-median functional prevalence.

## Remarks

Although the shape of the sole impression MZM Ge33302 is less mediolaterally expanded, the significantly long pedal digit V with the pV/pIV ratio of 0.88, the medial-median functional prevalence of the pes imprint, and the less delimited broad palm impression support the assignment of the MZM Ge33302 to *I. sphaerodactylum*. The ichnotaxon *Ichniotherium praesidentis* differs from MZM Ge33302 based on the presence of a prominent manual basal pad I impression, an inversed alternation of the pattern of manus-pes couples, a strongly outward rotation of the pes imprints, a lower stride length/pes length ratio, and a more acute pace angulation (Fig. 4A; Material SI). In turn, *Ichniotherium cottae* differs from MZM Ge33302 in having a substantially shorter pedal digit V impressions (pV/pIV ≤ 0.60), less extensive well-defined sole/palm impressions that are often clearly separated from the digit imprints, and usually higher stride length/pes length ratio and more obtuse pace angulation (Figs. 4C–4F; Material SI).

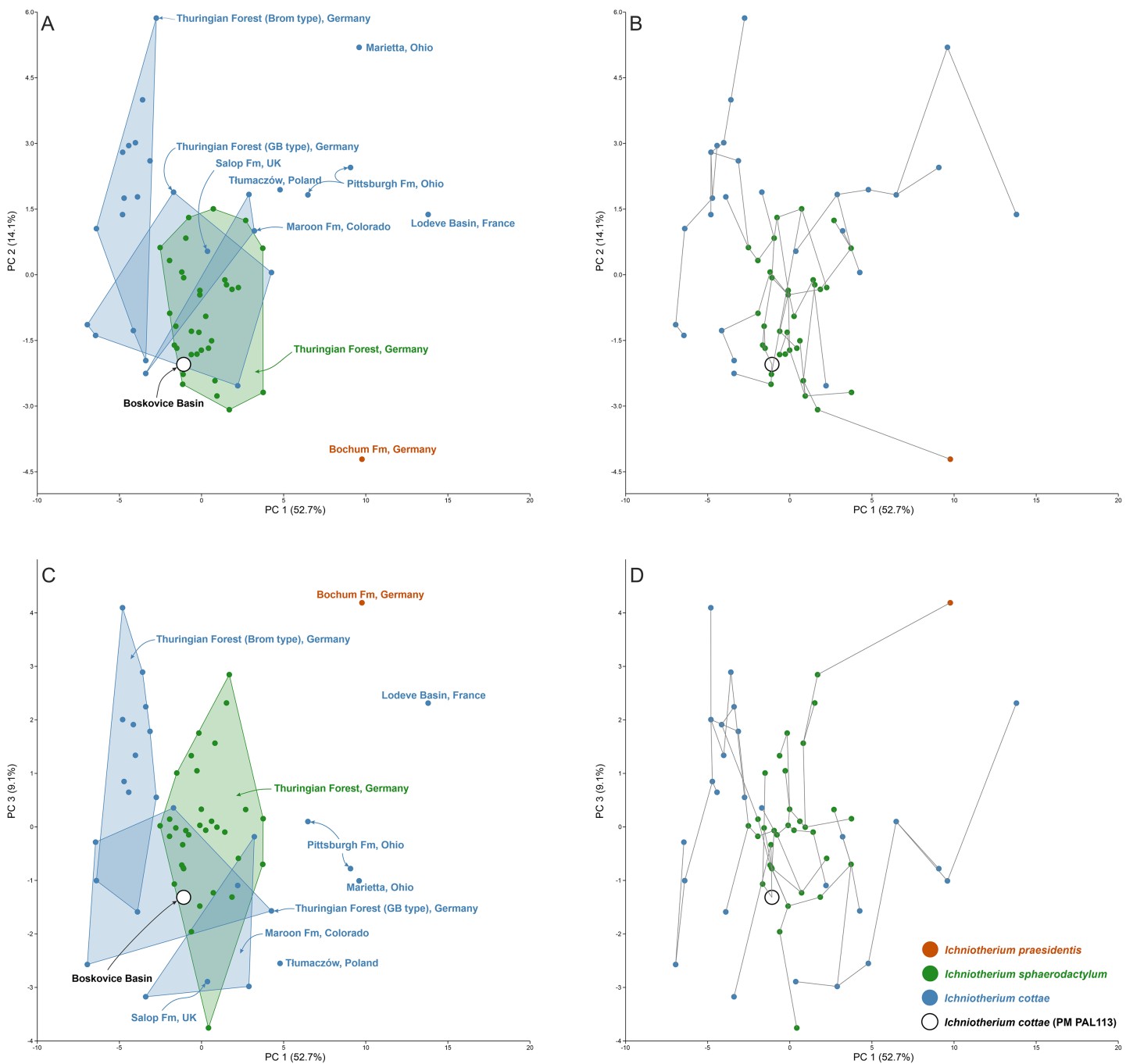

**Figure 5 Results of the principal component analysis (PCA).** The morphospace occupation of *Ichniotherium* trackways (A, C) and the minimum spanning tree (B, D) along PC 1 *vs* PC2 (A, B) and along PC1 *vs* PC3 (C, D). The analysis was performed using z-transformed values of Sp, Sm, Pp, Pm, Dpm, Gp, Gm, GAD, Pap, Pam, Oap, Oam, pI–pV (five values), mI–mV (five values), pL, mL, Sp/pL, Pp/pL, Gp/pL, (Gp-Gm)/pL, GAD/pL, Dpm/pL, and Oam-Oap (Material SI; see the main text for abbreviations).

## Results of the principal component analysis and box plots

The results of the principal component analysis (PCA) show broadly overlapping morphospace occupation of trackways assigned to *I. cottae* and *I. sphaerodactylum* that are concentrated near the centre of the biplots and widely separated from a trackway assigned

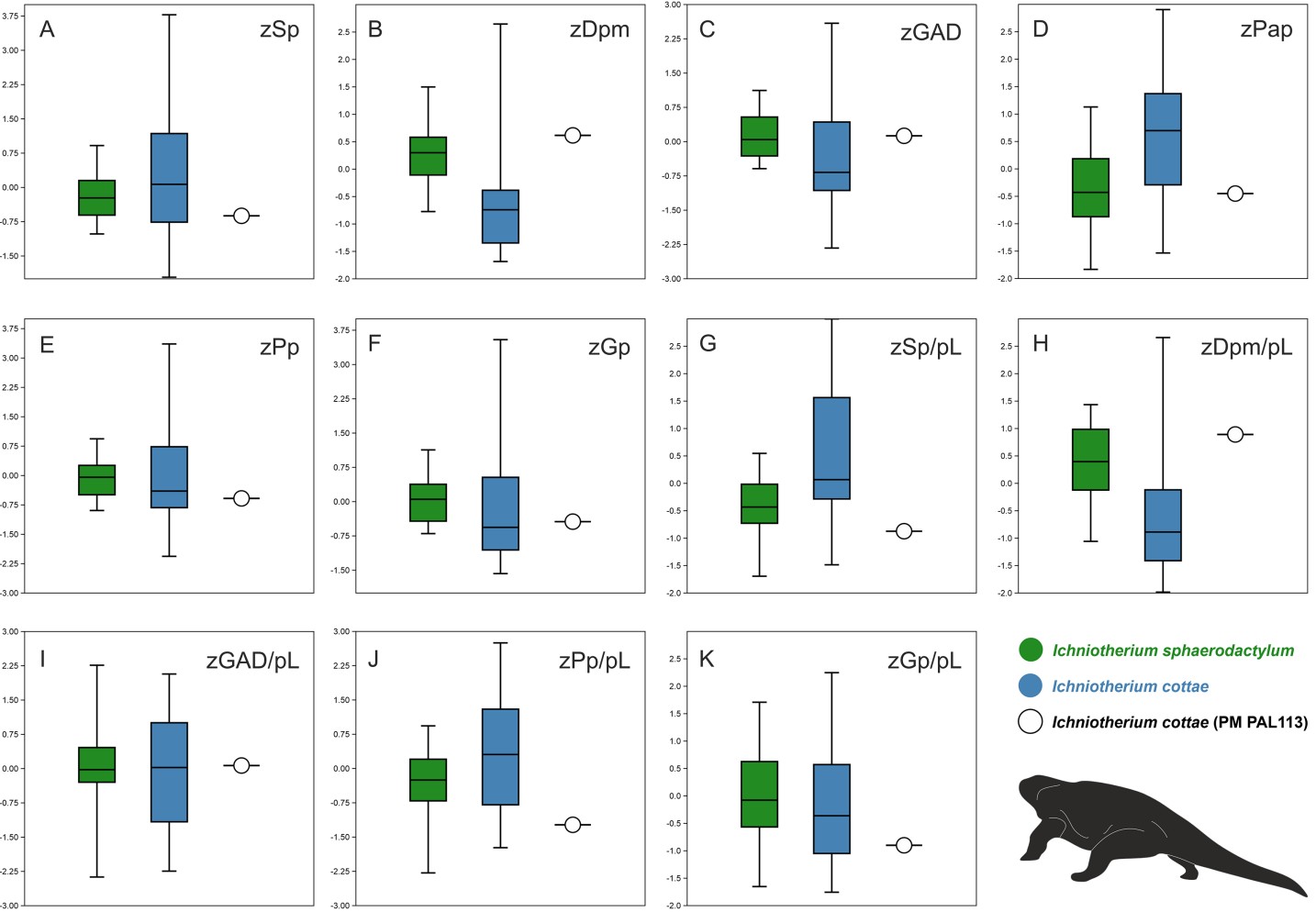

**Figure 6  Box plots.** Box plots showing comparisons of ranges of z-transformed values (prefix 'z') of selected parameters measured for tracks and trackways of *I. cottae*, *I. sphaerodactylum*, and PM PAL113 (Material SI; see the main text for abbreviations).

to *I. praesidentis*, which reflects the unique morphology and pattern of that ichnotaxon (*Voigt & Ganzelewski, 2010*).

In the biplots illustrating the highest percentage of variance (PC1 *vs* PC2 up to PC1 *vs* PC 5; PC1 (52.7%), PC2 (14.1%), PC3 (9.1%), PC4 (6.6%), and PC5 (4.4%)), PM PAL113 is placed near the center of the plots; on the negative sides of the axes and near or at the overlap of the *I. sphaerodactylum* morphospace and *I. cottae* from the "Gottlob-Birkeide type" (Fig. 5; Material SI). The minimum spanning tree additionally shows that PM PAL113 connects with *I. sphaerodactylum* specimens. For detailed values behind the plots and extended results of the PCA, see Material SI.

Comparisons of ranges of z-transformed values of selected parameters measured for tracks and trackways described as *I. cottae* and *I. sphaerodactylum* with those obtained from PM PAL113 further show that the values of these two ichnotaxa often overlap (Figs. 6A–6K). Nevertheless, a distinct variation can be observed in the distance between successive pes and manus imprints (Dpm) and Dpm/pes length ratio, pedal pace

A                B                C

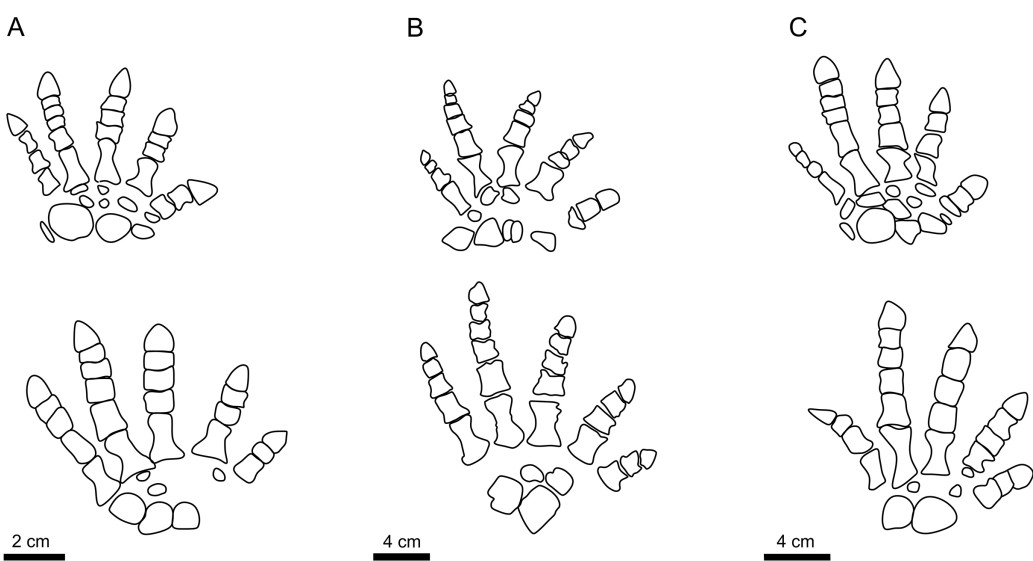

2 cm                 4 cm                 4 cm

**Figure 7 The autopodia of diadectomorphs.** The autopodia of *Orobates pabsti* (A), *Limnoscelis paludis* (B), and *Diadectes absitus* (C). Illustrations follow *Voigt, Berman & Henrici (2007)*, *Kennedy (2010)*, and *Mujal & Marchetti (2020)*.   

angulation (Pap), and pedal stride length(Sp)/pes length ratio. In these three parameters, PM PAL113 shows values closer to those typically obtained from trackways referred to as *I. sphaeordactylum* (Figs. 6B, 6D, 6G, 6H).

## DISCUSSION AND CONCLUSION

### Trackway pattern and functional implications

Specimen PM PAL113 shows an intriguing combination of features, characterized by presence of track parameters diagnostic for *I. cottae* (see above) and a trackway pattern with a high manus-pes distance/pes length ratio, a lower stride length/pes length ratio, and a lower manual and pedal pace angulation (Figs. 2, 4H; Material SI). Such features are typically observable in trackways attributed to *I. sphaerodactylum* (Figs. 4B; Material SI), that are usually interpreted to result from a relatively longer and more flexible trunk of the trackmakers and a more pronounced sprawling posture that allows a shorter stride and thus a lower maximum speed of walking (see *Buchwitz & Voigt, 2018*). It is worth noting, however, that the high degree of sprawling was questioned in the studies by *Nyakatura et al. (2015, 2019)* that explored the locomotion, body mass, and joint mobility of *Orobates pabsti* (using a 3D skeletal reconstruction and *I. sphaerodactylum* tracks), concluding that the *Orobates* movement was relatively erect, balanced, and mechanically power-saving in comparison to earlier tetrapods (*Nyakatura et al., 2019*).

The higher distance between the successive manus and pes imprints has been observed in several other specimens attributed to *I. cottae*; these include a specimen from the Gzhelian (uppermost Carboniferous) of the Pittsburgh Formation, Ohio (see *Buchwitz et al., 2021*), included in the "Hainesi-Willsi type" by *Buchwitz & Voigt (2018)*; specimen from the Asselian–Sakmarian (lower Permian) of the Gottlob and Birkheide locality in Thuringia, Germany, included in the "Gottlob-Birkheide type" by *Buchwitz & Voigt*

(2018); and specimen from the Asselian (lowermost Permian) of the Lunas locality, France (*Mujal & Marchetti, 2020*). However, the late Carboniferous "Hainesi-Willsi type" of *I. cottae* shows a distinct outward rotation of the pedal imprints (Figs. 4C, 4D) which occurs also in older *Ichniotherium* tracks referred to as *I. praesidentis* (Fig. 4A) that are Moscovian (middle late Carboniferous) in age (*Buchwitz & Voigt, 2018*). Thus, it differs significantly from PM PAL113 as well as the majority of younger *Ichniotherium* ichnotaxa (see Figs. 4B, 4E–4H).

A gradual change of the manus and pes imprint orientation in the *Ichniotherium* trackways has already been observed by *Buchwitz & Voigt (2018)* who provided detailed description of the evolution of diadectomorph locomotion based on the succession of trackmakers of *I. praesidentis*, *I. sphaerodactylum*, and three morphotypes of *I. cottae*, which were distinguished by *Buchwitz & Voigt (2018)*; the "Hainesi-Willsi type" (Moscovian–Gzhelian; upper Carboniferous; Figs. 4C, 4D), the "Birkheide-Gottlob type" (Asselian–Sakmarian; lower Permian; Figs. 4E, 4F), and the "Bromacker type" (Sakmarian–Artinskian; lower Permian; Fig. 4G). The appearance of these trackways has been interpreted to exhibit a tendency towards trunk shortening combined with decreasing of its flexibility, narrowing of the sprawling posture, and progressively inward orientation of the manus in the touch-down phase. Such modifications resulted in a higher maximum speed of walking and a higher efficiency of land movement. The youngest "Bromacker type" with its typically 'complete' overstepping of the manus-pes couples, the most pronounced inward rotation of the manus and the pes impressions, most obtuse pace angulation, narrowest gauge, and significantly higher pace and stride lengths (Fig. 4G) differs from all of older *Ichniotherium* trackways, including PM PAL113 (Fig. 4H). The characters observed in the tracks and trackways attributed to *I. cottae* additionally indicate that the later phase of the diadectomorph evolutionary history was characterized by changes in limb posture and joint mobility rather than by significant differences in trunk proportion (see *Buchwitz & Voigt, 2018*).

## The identity of the trackmaker of PM PAL113

*Buchwitz & Voigt (2018)* interpreted the morphotypes of *I. cottae* to have been registered by later-diverging representatives of Diadectidae, such as *Desmatodon*, *Diasparactus*, and *Diadectes*. In turn, the nearly complete skeleton of *Diadectes absitus* found at Bromacker was used to correlate the taxon with the "Bromacker type" of *I. cottae* (see *Voigt, Berman & Henrici, 2007*). The trackmaker of PM PAL113 shows a slightly outward to parallel-oriented pedal imprints, parallel to slightly inward-oriented manual imprints, relatively short length of stride and pace, lower pace angulation but narrower gauge and short pedal digits V (Material SI; Figs. 2, 4H). These features are closer to those observed in the "Birkheide-Gottlob type" (Figs. 4E, 4F and 5) which may be associated with *Diadectes*-line diadectids (see *Buchwitz & Voigt, 2018*, fig. 18). However, PM PAL113 has distinctly less inwardly-rotated manus and pes imprints and higher manus-pes distance/pes length ratio (Material SI). Such condition is fundamentally distinct from that typically observed in trackways registered by *Diadectes* (*e.g.*, the "Bromacker type"; Fig. 4G) and instead resembles most of the trackways referred to as *I. sphaerodactylum* (Fig. 4B) which are

commonly attributed to earlier-diverging diadectids, such as *Orobates* found at the Bromacker locality (*Voigt, Berman & Henrici, 2007*; *Buchwitz & Voigt, 2018*), or perhaps early-diverging diadectomorphs in general, such as *Limnoscelis*, which shares with *Orobates* the same phalangeal formula (2-3-4-5-3 for the manus and 2-3-4-5-4 for the pes (Figs. 7A, 7B)) and differs from *Diadectes* that shows the same phalangeal formula (2-3-4-5-3 (Fig. 7C)) in both, the manus and the pes (*Voigt, Berman & Henrici, 2007*; *Kennedy, 2010*). These taxa are characterized by a slightly higher number of presacral vertebrae (21 in *Diadectes absitus*, 23 in *Limnoscelis paludis*, 26 in *Orobates pabsti* (*Berman & Henrici, 2003*; *Voigt, Berman & Henrici, 2007*; *Kennedy, 2010*)) and, thus, more elongated trunks. The combination of features captured in PM PAL113 is also well depicted through the results of our PCA (PC1 *vs* PC2 up to PC1 *vs* PC5) where the specimen fall within the overlap of the morphospaces occupied by *I. cottae* and *I. sphaerodactylum* tracks (Figs. 5, 6). Owing to the fact that the propodial-to-epipodial proportions in *Orobates* and *Diadectes* were nearly identical (see *Voigt, Berman & Henrici, 2007*), the differences in diadectomorph trackway pattern might additionally stem, as already mentioned above, from differing body mass and limb joint mobility which largely determine the mode of locomotion (see *i.e.*, *Nyakatura et al., 2015*, *2019*) and which were also most likely reflected in the trackway pattern of the PM PAL113.

### The significance of MZM Ge 33302 and the identity of its trackmaker

The manus-pes couple of MZM Ge33302 (Fig. 3) with the relatively longer pedal digit V (pV/pIV ratio = 0.88) attributed to *I. sphaerodactylum* represents only the second specimen of this ichnotaxon from the European part of Pangea. Owing to the morphology of the tracks, the manus-pes couple has been likely registered by an *Orobates*-like or perhaps a *Limnoscelis*-like taxon. Interestingly, *Limnoscelis* was also associated with tracks from the Asselian of the Lunas locality in France that were referred to *I. cottae*. This association has been based on a trackway pattern similar to that of the "Hainesi-Willsi type" or *I. praesidentis* and a strong medial functionality of the Lunas pes imprints (*Mujal & Marchetti, 2020*). The Lunas pes imprints show a high pV/pIV ratio as well. However, this most likely stemmed from an error in the obtained measurements, caused by poor preservation of the tracks (E. Mujal, 2023, personal communication).

Although it is impossible to associate a single manus-pes couple with a certain diadectomorph taxon, especially without any associated skeletal record in the Boskovice Basin, MZM Ge 33302 has been clearly registered by an earlier-diverging member of the clade than PM PAL113. Thus, there have been at least two distinct diadectomorphs in the Asselian (earliest Permian) equatorial ecosystems of what is today the Boskovice Basin in the Czech Republic.

## ACKNOWLEDGEMENTS

We are indebted to Richard Knecht and Josef Zacpal (both Podhorácké Museum in Předkláštěří, Czech Republic) for access to PM PAL113, Tomáš Viktorýn (Brno, Czech Republic) for his help during fieldwork at Zbýšov, which led to the discovery of MZM Ge33302, Eudald G. Mujal (State Museum of Natural History Stuttgart, Germany) for

discussion on diadectomorph footprints from Zbýšov and Lunas, Jakub Březina (MZM) for discussion on the lithostratigraphic location of the trackway-bearing fossil sites, and Martin Hanáček (Vlastivědné museum Jesenicka, Czech Republic) for discussion on the genesis of clastic sediments at these sites. Finally, we would like to extend our gratitude to Graciela Piñeiro (Universidad de la República, Uruguay) for handling our manuscript, and Michael Buchwitz (Museum für Naturkunde Magdeburg, Germany) and Heitor Francischini (Universidade Federal do Rio Grande do Sul, Brazil) for their constructive reviews.

## INSTITUTIONAL ABBREVIATIONS

**MZM**     Moravian Museum, Brno, Czech Republic
**PM**      Podhorácké Museum, Předklášteří, Czech Republic

## OTHER ABBREVIATIONS

**Dpm**     distance between successive pes and manus imprints
**GAD**     glenoacetabular distance
**Gm**      manual gauge width
**Gp**      pedal gauge width
**mI–V**    length of manual digits I–V
**mL**      manus length
**mW**      manus width
**Pam**     pes angulation for the manus imprints
**Pap**     pace angulation for the pes imprints
**pI–V**    length of pedal digits I–V
**pL**      pes length
**pW**      pes width
**Pm**      manual pace length
**Pp**      pedal pace length
**Sm**      manual stride length
**Sp**      pedal stride length
**Oam**     orientation of the manus imprints
**Oap**     orientation of the pes imprints with respect to the direction of movement (positive values reflect inward rotation, negative values reflect outward rotation)

### Funding

This study was funded through the institutional support of long-term conceptual development of research institutions provided by the Ministry of Culture of the Czech Republic (ref. MK000094862) and the project MUNI/A/1261/2022 of the Faculty of Science at the Masaryk University in Brno, Czech Republic (Martin Ivanov). The funders

had no role in study design, data collection and analysis, decision to publish, or preparation of the manuscript.

## Grant Disclosures
The following grant information was disclosed by the authors:
Ministry of Culture of the Czech Republic: MK000094862.
Project MUNI/A/1261/2022 of the Faculty of Science at the Masaryk University in Brno, Czech Republic (Martin Ivanov).

## Competing Interests
The authors declare that they have no competing interests.

## Author Contributions
- Gabriela Calábková conceived and designed the experiments, performed the experiments, analyzed the data, prepared figures and/or tables, authored or reviewed drafts of the article, and approved the final draft.
- Daniel Madzia conceived and designed the experiments, performed the experiments, analyzed the data, prepared figures and/or tables, authored or reviewed drafts of the article, and approved the final draft.
- Vojtěch Nosek analyzed the data, authored or reviewed drafts of the article, and approved the final draft.
- Martin Ivanov analyzed the data, authored or reviewed drafts of the article, and approved the final draft.

## Data Availability
  The raw data is available in the Supplemental Files.

## Supplemental Information
Supplemental information for this article can be found online at http://dx.doi.org/10.7717/peerj.16603#supplemental-information.

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
