# Peer review of "Tracking ‘transitional’ diadectomorphs in the earliest Permian of equatorial Pangea"

_PeerJ, doi:10.7717/peerj.16603_

## Round 0.1 · original submission · Minor Revisions

Dear authors,

I am pleased to inform you that we now have two review reports about your manuscript entitled “Tracking ‘transitional’ diadectomorphs in the earliest Permian of equatorial Pangea”. Both reviewers considered that your article is well written and your results are mostly convincing. However, there are some little concerns marked by the reviewers that require your attention and they should be addressed. Thus please, consider revising all the comments and recommendations from the reviewers before submitting a new version of your manuscript. Otherwise, in case you disagree with some of them, you can explain your points in the rebuttal letter.
Please, note that one of the reviewers also submitted an annotated pdf file, which also needs your consideration.

My recommendation is that you include a brief Geological Setting paragraph in order to facilitate checking of the evidence that placed the involved lithostratigraphic units in the earliest Permian. This will also facilitate the access of the readers to this relevant information, given that you are describing two new ichnotaxa that will enhance the fossil record of these strata.

I hope you will find the reviewers' recommendations useful for improving your manuscript, allowing you to submit the updated version very soon.
Best wishes,
Graciela Piñeiro

·

Basic reporting

No comment.

Experimental design

No comment.

Validity of the findings

No comment.

Additional comments

No comment.

·

Basic reporting

The manuscript by Calabkova and colleagues on is well written and organized; the relevant studies have been sufficiently covered. The material of footprints and a trackway from the Boskovice Basin is adequately assigned to the ichnospecies Ichniotherium cottae and I. sphaerodactylum and well worth publishing.

Experimental design

The data, pictures, supplement look okay, methodology is sufficiently described and in agreement with common standards; regarding the multivariate analyses: I wonder how the newly described trackway would have been classified by certain algorithms (e.g. according to linear discriminant functions/LDA based on the previous sample).

Validity of the findings

I would agree with the authors’ observation that the trackway pattern of the I. cottae trackway differs from typical trackways assigned to this ichnotaxon – difficult to tell whether this pattern is transitional or not based on a single trackway but arguably individual variables – especially normalized/standardized glenoacetabular distance (“apparent trunk length”), its relation to stride length, gauge width and pace angulation should be depicted as individual schemes (e.g. box plots, bivariate plots) and addressed in some detail – in addition to their consideration in multivariate analyses. Please name in the figure captions for Fig 3 the variables/parameters which were included in the PCA.
There is also an alternative interpretation - that the I. cottae trackway producer was not much different from other I. cottae trackmakers (described by Buchwitz and Voigt and others) in terms of body proportions, but that the gait (i.e. limb phase) was different, causing the diverging positions of manus and pes imprints along the track in the Czech specimen. You may discuss the material described by Francischini et al. (2019) in this context – the specimen from the Coconino sandstone has a trackway pattern distinct from normal Ichniotherium (short steps) which is not necessarily due to anatomical differences but might have had other causes (e.g. walking on a slope/an inclined plane) which affected, among others, the coupling value.
I find the authors’ interpretation plausible that the early occurrence of I. sphaerodactylum (probably older than the other I. sphaerodactylum samples from Germany, North America) indicates that this kind of foot shape/proportion was not only produced by Orobates but also earlier representatives of the Diadectomorpha.

Additional comments

L142 Systematic paleoichnology: “Diadectomorpha Watson 1917” should be deleted here because it does not represent an ichnotaxon and should merely be mentioned later in the discussion, e.g. under “(likely) producer groups”
L186 “sphaerodactylum” instead “sphaerodatylum”
L273 “absitus” instead of “abitus”

---

## Round 0.2 · Minor Revisions

Dear authors,


I apologize for the delay in my decision; last week was very busy for me.
Well, the revised manuscript that you sent is undoubtedly much better than the already good submitted initially. Thanks for taking into account most of the reviewers’ suggestions and recommendations. Also, the inclusion of a Geological Setting section was very helpful, thanks.

I have only a minor concern about the decision you followed with respect to one comment from Reviewer 2, Dr. Michael Buchwitz regarding considering a slight deviation from the pattern of I cottae and then considering the isolated manus-pes couple found as the presence of two different ichnotaxa.

As far as I know ichnofossils determine behavior or an action, and the presence of two ichnotaxa does not means that you infer the presence of two diadectomorph species. Mainly if you do not find other evidences as for instance, associate bones. Therefore, if you found a little morphological difference within a general particular pattern or morphotype, it is possible to consider that the tracks were produced by, for instance, ontogenetically older or younger individuals? And it was not clear for me your argument to dismiss what the reviewer suggested about that the tracks could have been produced over a distinct substrate conditions. Perhaps it could be good to provide a new figure including the manus-pes couples attributed to I. sphaerodactylum and the same for I. cottae, indicating the main morphological differences that you considered as enough to recognize the presence of two ichnotaxa, that cannot be explained by substrate influence.

Furthermore, include in the discussion the results of your principal component analysis (PCA) showed that I. cottae and I. sphaerodactylum broadly overlap and I. praesidentis occupates a different morphospace, and explain the implicances of these results in your conviction that there is reason to describe two track ichnospecies.

The information about the stratigraphic distribution of the ichnotaxa was not very clear for me, because it seems that the specimen assigned to I. sphaerodactylum was indeed found at a transitional sequence from another locality that possibly preserves a section which is interpreted as a little stratigraphically below that which provided the specimen assigned to I. cottae. If my observations are correct, please clarify in that way to the readers, particularly to people not familiar with the geology of the studied region. Perhaps to include a figure showing a simplified and integrated stratigraphic section of both the localities that provided the fossil traks would be useful.

Thus, in sum, I expect that you prove your hypotheses about the presence of two track ichnotaxa that under your consideration, may represent two different diadectomorphs species, by providing more convincing evidence: 1) a figure contrasting the eventual two ichnotaxa and 2) other figure showing a simplified and integrated stratigraphic section indicating the position at which the studied specimens were found.

Hope to see your revised manuscript resubmitted very soon.

Best regards,
Graciela Piñeiro

---

## Round 0.3 · Minor Revisions

Dear authors,
Thanks for your rebuttal. Indeed I based my comments mainly in the text showed below that was included in the Dr. Buchwitz review. Thus, there was not any misunderstanding here.

I would agree with the authors’ observation that the trackway pattern of the I. cottae trackway differs from typical trackways assigned to this ichnotaxon – difficult to tell whether this pattern is transitional or not based on a single trackway but arguably individual variables – especially normalized/standardized glenoacetabular distance (“apparent trunk length”), its relation to stride length, gauge width and pace angulation should be depicted as individual schemes (e.g. box plots, bivariate plots) and addressed in some detail – in addition to their consideration in multivariate analyses. Please name in the figure captions for Fig 3 the variables/parameters which were included in the PCA.

There is also an alternative interpretation - that the I. cottae trackway producer was not much different from other I. cottae trackmakers (described by Buchwitz and Voigt and others) in terms of body proportions, but that the gait (i.e. limb phase) was different, causing the diverging positions of manus and pes imprints along the track in the Czech specimen. You may discuss the material described by Francischini et al. (2019) in this context – the specimen from the Coconino sandstone has a trackway pattern distinct from normal Ichniotherium (short steps) which is not necessarily due to anatomical differences but might have had other causes (e.g. walking on a slope/an inclined plane) which affected, among others, the coupling value.

I am not specialized in ichnology, but because I am handling this article, I have the function to make it understandable to colleagues that also are not dedicated to vertebrate ichnology, but evidence of the presence of some taxa through the foot prints in geological sections is relevant to their researching. That said, in my concept, the reviewer is suggesting an alternative hypothesis to your results, which introduces certain doubt about if the differences that you described are indicating a transitional morphology or they are related to “other causes (e.g. walking on a slope/an inclined plane)”. That means to me that the reviewer is marking that a different interpretation is possible.

Therefore, whatever the reason for the differences is, I suggested adding a figure to show better the differences between the Czech specimen and other I. cottae specimens. Perhaps in the field where you work, you would consider that it is not necessary, because your article will be read only by ichnologists working on these ichnotaxa. But PeerJ is a journal with a wide scope.

Concerning the geological setting I also found that the description of the stratigraphic position of both the ichnotaxa was not enough clear and that would be very easy for you to include a figure showing the original setting of the tracks in the corresponding lithostratigraphic units. I am gratified that you decided to accept the request and add a nice Figure 1 that clearly shows that the “fine-grained sandy floodplain deposits” belong indeed to different formations.

In sum, for me to be satisfied with the content of your manuscript to be published in PeerJ, I would prefer that you be more nuanced in your interpretation and more clearly designate the discrepancies between the different traces assigned to this ichnospecies and trackmakers. The main aspects raised in their rebuttal concerning these points should be included in the manuscript for the sake of complete reproducibility. The inclusion of the already requested comparative figure still is recommendable.

With my best regards,
Graciela Piñeiro

PS. I wish to apologize for having sent the wrong annotated manuscript, indeed it only explained in more detail the reasons for which I requested the new figures. I think that my mistake was due to the fact that I have three folders about diadectomorphs with edited manuscripts for PeerJ. There is no reason to include it now. I am very sorry.

---

## Round 0.4 · accepted · Accept

Dear authors,

Thank you very much for having decided to include what is now Figure 4. Indeed, it demonstrated that I was right when I perceived that such a figure comparing the trackway morphotypes in discussion, would help readers to better understand the conclusions to which you arrived at the end. Furthermore, I am glad to see that you made some changes to the main text in order to provide additional information supporting your results.

For the mentioned reasons, I am taking the final decision for your manuscript.

Congratulations!
Best regards,
Graciela Piñeiro